

# 2 Surface elevation and ice thickness data between 2012 and

# 3 the ablation area of Artesonraju Glacier, Cordillera Blanca, Perú.

Jonathan Oberreuter[1], Edwin Badillo-Rivera[2,3], Edwin Loarte[2], Katy Medina[2], Alejo Cochachin[4], José
Uribe[1]
[1]Centro de Estudios Científicos (CECs), Valdivia, Chile.
[2]Instituto Nacional de Investigación en Glaciares y Ecosistemas de Montaña (INAIGEM), Huaraz, Perú.
[3]Facultad de Ingeniería Ambiental y Recursos Naturales, Universidad Nacional de Callao, Bellavista-Callao, Perú.
[4]Área de Evaluación de Glaciares y Lagunas, Autoridad Nacional de Agua (ANA), Huaraz, Perú.
*Correspondence to*: Jonathan Oberreuter (jober@cecs.cl)
**Abstract.** We present a representative set of data of interpreted ice thickness and ice surface elevation at the ablation area of
the Artesonraju glacier between 2012 and 2020. The ice thickness was obtained by means of Ground Penetrating Radar
(GPR), while the surface elevation was by means of automated total stations and mass balance stakes. The data coverage is
about 14% of the whole glacier area. The results from GPR data show a maximum depth of 235±18 m and a decreasing
mean depth ranging from 134±18 m in 2013 to 110±18 m in 2020. Additionally, we estimate a mean ice thickness change
rate of -4.2±3.2 m yr$^{-1}$ between 2014 and 2020 with GPR data alone, which is in agreement with the elevation change in the
same period. The latter was estimated with the more accurate surface elevation data, yielding a change rate of -3.2±0.2 myr$^{-1}$,
and hence, confirming a negative glacier mass balance. The data set can be valuable for further analysis when combined with
other data types, and as input for glacier dynamics modeling, ice volume estimations, and GLOF (glacial lake outburst flood)
risk assessment. The complete dataset is available at https://doi.org/10.5281/zenodo.5571081 (Oberreuter et al, 2021).
**1 Introduction**
The glacier variations are sensitive to climate change (Oerlemans, 2005), particularly that of tropical glaciers (Rabatel et al.,
2013; Vuille et al., 2008) located between 23.43°N and 23.43°S, whose ablation processes occurs during the entire year,
while accumulation only during the austral summer (Gonzales Molina & Vacher, 2014). Hence, any variation in climate
affects tropical glaciers faster than glaciers from different latitudes (Uani, 2018). The tropical region of the Andes is
experiencing glacier mass loss since the end of Little Ice Age and has been accelerating since 1980 (Rabatel et al., 2013;
Unai, 2018), probably due to increase in air and ground temperatures. Vuille et al. (2015) showed an increase rate in air
temperature of 0.13°C per decade in the last 60 years, while Aguilar-Lome et al. (2019) showed that the ground temperature





increased at a rate of 0.17°C per decade above 5000 m.a.s.l., causing tropical glaciers to be the type with the most area
shrinkage globally at a rate of 1.6% per year (Li et al., 2019).
The tropical Andes concentrate ~99% of the tropical glaciers of the planet (Kaser & Osmaston, 2002), and of these, ~68%
are located in Perú (Veettil & Kamp, 2019), distributed in the 18 glacier cordilleras. The Artesonraju glacier is located in the
larger tropical glacier chain in the world: Cordillera Blanca, which concentrates ~40% of the Peruvian glacier surface, that is,
448.81 km$^2$ (INAIGEM, 2017), has a total length of 247 km (INAIGEM, 2018b) and it's simultaneously the cordillera with
the greater area glacier loss: 277.5 km$^2$ in the last 54 years (INAIGEM, 2018b). Also, it contributes ~40% of water supply to
the basin during dry season (Mark et al., 2005).
The glacier surface and volume reduction is in some cases associated with the generation of proglacial lakes and vice versa.
This association is stronger than that with land-terminating glaciers (King et al., 2018). Artesonraju glacier is one of those
cases, with an area reduction of 10% in 46 years, from 5.97 km$^2$ in 1970 (Ames, 1988) to 5.43 km$^2$ in 2016 (INAIGEM,
2018a). Its front has retreated 133 m in the period 2006-2019 (~10.2 m yr$^{-1}$), causing a severe expansion of the Artesoncocha
Alta lake, from 2,020 m$^2$ in 2003 to 22,314 m$^2$ in 2015 (INAIGEM, 2016).
In order to estimate the glacier ice thickness and its variations, several methods can be applied: ice-core drilling (Garzonio et
al., 2018; Zagorodnov et al., 2005), ice thickness modelling by means of: a) glacier morphology and slope (Campos, 2020;
Frey et al., 2014; Helfricht et al., 2019; Huss & Farinotti, 2012; Paul & Linsbauer, 2012), b) glacier surface velocities
(Gantayat et al., 2014; Sattar et al., 2019); and ice thickness measurement by means of geophysical methods such as seismics
and GPR (Booth et al., 2013; Colombero et al., 2019; Shean & Marchant, 2010; Zhao et al., 2016). GPR is a non-invasive
technique based on transmission and reception of radiowaves ranging normally from 10 MHz to 6 GHz (Zhao et al., 2016;
De Pascale et al., 2008), which has been successfully utilized in mountain glaciers for estimating ice thickness (Gacitúa et
al., 2015; Grab et al., 2021; Singh et al., 2010; Liu et al., 2020; Bohleber et al., 2017; Santin et al., 2019). However, up to
date, the GPR technique has been scarcely applied in tropical glaciers in Perú and the records of systematic monitoring in
Peruvian glaciers are also sparse. Hence, this study aims to contribute with ice thickness data and ice surface elevation data
at Artesonraju Glacier in Cordillera Blanca (Caraz-Áncash), from 2012 to 2020, which is useful for ice volume estimates and
as input for glacial dynamic models.
**1.1 Study area**
The study area comprises the Artesonraju glacier (Figure 1), a tropical glacier located in the northern side of the Cordillera
Blanca. According to the most recent national glacier inventory, its centroid is located at 8°57'29'' S, 77°38'0'' W, its
elevation ranges from 4902 to 5675 m.a.s.l. and an area of 5.43 km$^2$ (INAIGEM, 2016, 2018a). The ablation area of the
glacier consists of two parts: an uncovered part and a debris-covered part.



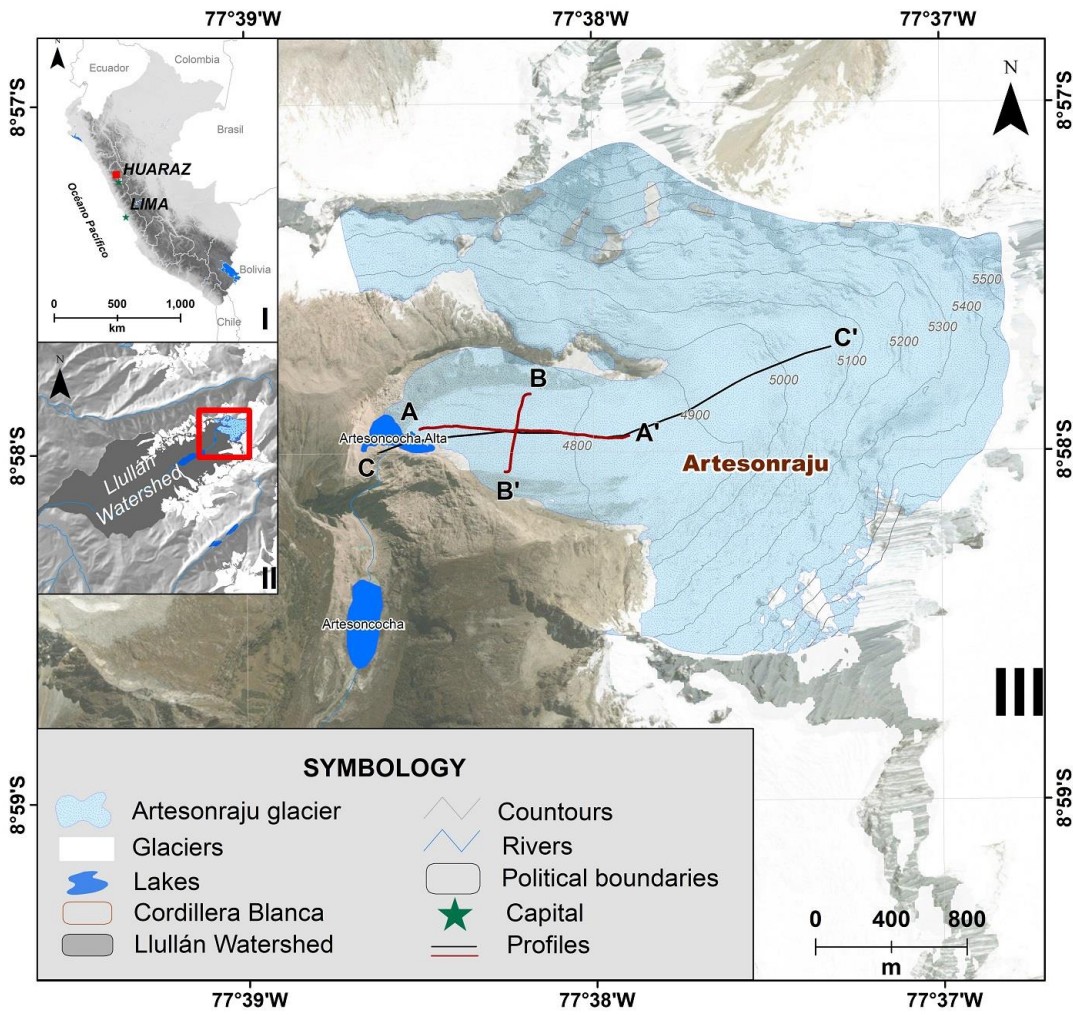

**Figure 1 Study area of Artesonraju glacier. Glacier boundary from INAIGEM (2018a).**

## 2 Data and methods

### 2.1 GPR data coverage

The GPR data coverage from 2013 to 2020 is described in Figure 2 and in Table 1. Data from years 2013-2017 were collected by the Autoridad Nacional de Agua (ANA), while data from years 2018-2020 were collected by Instituto Nacional de Investigación en Glaciares y Ecosistemas de Montaña (INAIGEM). All data were obtained with the same radar system described in section 2.2.





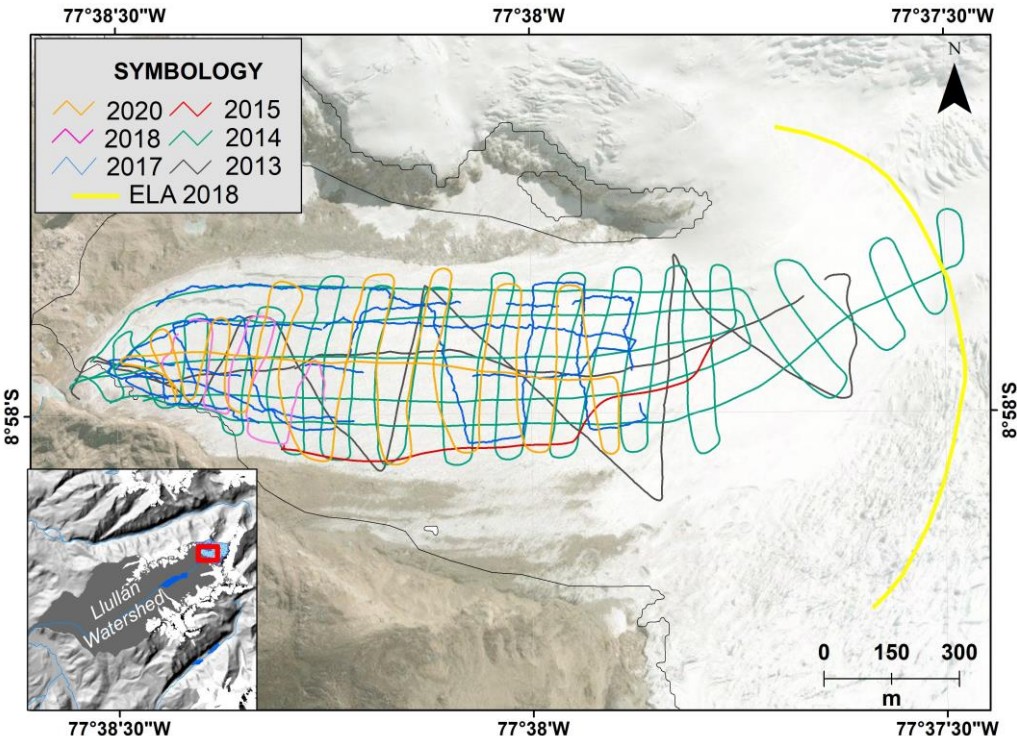


**Figure 2 GPR measurements coverage at Artesonraju glacier. Glacier boundary from INAIGEM (2018a). Background image:**
**Landsat, 4th of July, 2014.**
**Table 1  Ice thickness datasets used in this study.**

| Date | Collected by | Total  profile  length (km) | Coverage  area  as % of total  area  (5.43 km$^2$) |
|---|---|---|---|
| 2013-07-11 | ANA | 5.3 | 12 |
| 2014-05-31/2014-06-03 | ANA | 18.6 | 14 |
| 2015-07-21 | ANA | 1.1 | 1 |
| 2017-05-30 | ANA | 8.0 | 7 |
| 2018-08-16 | INAIGEM | 1.4 | 1 |
| 2020-10-22 | INAIGEM | 6.8 | 7 |


**2.2 GPR System**
A low-frequency impulse-type GPR developed by Unmanned Industrial LTDA  has been used in order to estimate the ice
depth, similar to the one used by Bello et al (2020) to measure the ice thickness on King George island in Antarctica. The
GPR consists of a receiver, an impulse generator as transmitter, both with inbuilt GPS (Taoglas antenna, model AA.161) and



two bistatic-shaped antennas. Its main features are detailed in Table 2. The transmission unit is able to generate a high-voltage impulse at a pulse repetition frequency (PRF) of 1 kHz, with an output of 1.4 kV. Its two integrated GPS/GLONASS antennae (with GPS navigator accuracy) allow the trigger synchronization between receiver and transmitter, which is important for setting the beginning of each trace. The two antennas are dipole Wu-King type (Wu & King, 1965), with 5 MHz of central frequency and bandwidth. During the capture process, the raw data are transferred to a handheld rugged computer (PDA) for storing and real-time visualization of data. For a better deploying of the GPR, a group of five people was needed during the surveys as shown in Figure 3.

**Table 2  GPR features.**

| Features | Value |
|---|---|
| Dipole length (m) | 7 |
| Transmitter voltage (kV) | 1.4 |
| Central frequency (MHz) | 5 |
| Sampling rate (MHz) | 80 |
| Range resolution (m) | 16.8 |

The receiver unit digitalizes the signal at a sampling rate of 80 MHz with 16 bits of resolution, yielding a sample length (or time increment) of 12.5 ns, and is able to stack from 256 to 4096 traces, with a trace length from 256 to 1024 samples. The digital system enables the user to set a fixed trace length (in samples) between 256 and 1024 ($2^n$ format). The proper choice of the trace length depends on the estimated depth, ranging from 268 m to 1075 m, assuming a depth-averaged wave propagation velocity in ice (c) of 0.168 m/ns (Glen & Paren, 1975).

With the interpretation of the two-wave travel time ($TWTT$), we estimate the ice depth ($h$) using equation (1).

$$h = \frac{c * TWTT}{2} \qquad (1)$$

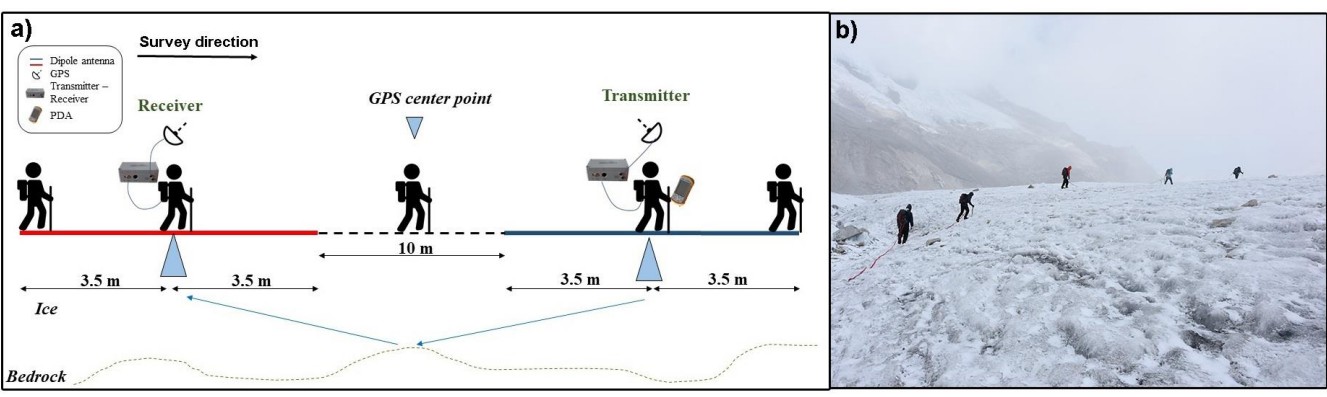

**Figure 3 a) GPR survey scheme performed in this study. The red and blue lines represent the dipole antennas of 7 m long each, with 10 m of spacing between the receiver and transmitter ends. b) GPR survey in the fieldwork at Artensonraju Glacier.**



## 2.2 GPR data processing

The first stage of the processing considers data preview and format conversion with the software RADAR View, which is the software provided by the GPR company. Then, the files are able to be imported into software Reflexw, where profiles are loaded and integrated with GPS data, and processing takes place, including the following steps:

a) Normalization of amplitude-saturated traces: Due to different sources of interference, traces are sometimes affected by saturation in the signal amplitude, which leads visually to discontinuities in the profile. In this case, the normalization consists of applying the calculation Amplitude/max(Amplitude) to every trace.

b) Dewow filter: Low-pass filter used to eliminate the "wow" effect, caused by a low frequency component in the signal. The filter is defined by a window length which should include the first arrival waveform. In this case, the window length parameter was set to 100 ns, which satisfies the requirement.

c) Butterworth bandpass filter: To reduce the noise that is present in the signal outside the frequency range, where the radar operates, which is centered at 5 MHz. The lower and higher boundaries of the filter were set to 2and 10 MHz, respectively.

d) Definition of zero time: The direct wave travelling through the air between source and receiver should be eliminated in order to estimate correctly the two wave travel times of the corresponding reflections. In this case, the zero time was equal for each dataset varying from 305 ns to 380 ns.

e) Trace interpolation: To have a better visualization of the profile and as a necessary step for the migration, an equidistant trace interpolation was performed. Here a 1 meter separation between traces was used, preserving a similar number of traces per profile, and consequently, keeping the same horizontal resolution.

f) Migration: In order to set the reflectors to their real position, a migration process was performed. It was used a 1D Kirchoff migration because of its better signal to noise ratio in this particular case. Here the migration profile window was set to 250 traces and the depth-average electromagnetic wave velocity was set to 0.168 m/ns. This velocity was in agreement with hyperbolae shapes.

g) Topographic correction: This procedure enables to move the traces up and down in the profile according to the elevation that is store in the trace header, which was defined when importing the GPS data to the profile The results enable to visualize correctly the surface elevation and the bedrock elevation along the profile (see lower panel of Figure 4).

After processing, the bedrock interface was manually picked at each radar profile and then exported to GIS software. The processed radargrams are also visualized in OpendTect in order to examine the crossover differences. An example of two processed profiles is shown in Figure 4. In general, there is good agreement in the crossover analysis (of the order of GPR error), as shown in the upper inset of the figure.



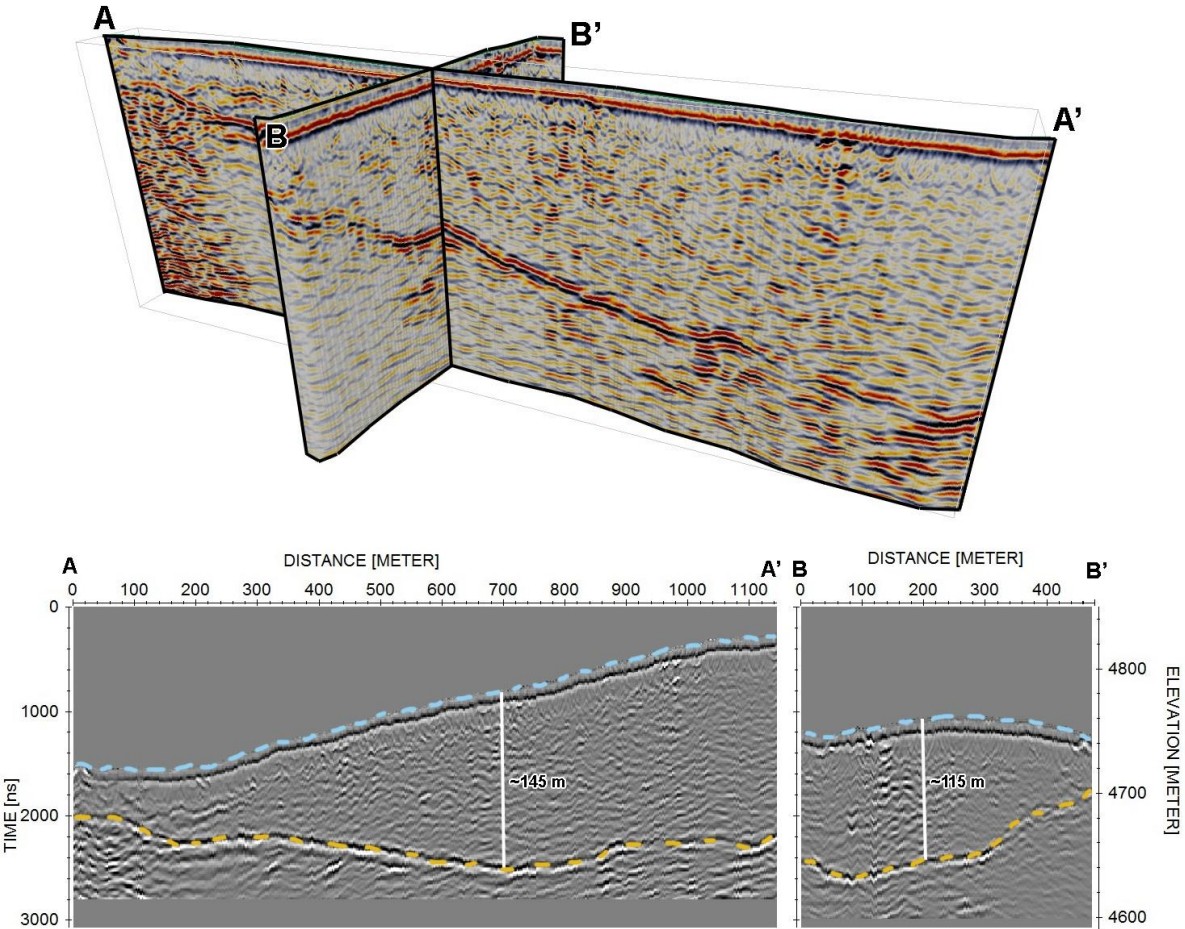

**Figure 4 Radargrams of Artesonraju glacier surveyed in 2020: profiles AA' and BB' as defined in Figure 1. Top panel shows a 3d view of longitudinal and transversal processed radar profiles (without topographic correction) and bottom panel shows the same processed profiles with topographic correction. The glacier surface is marked in color cyan and bedrock in yellow.**

## 2.3 Surface and bedrock elevation

Two sets of surface elevation data, which were surveyed by ANA, are considered in this paper: first, 2012, 2014, 2018 and 2020 Digital Elevation Models (DEM) obtained from topographic points; and second, 2014, 2015, 2018 and 2019 mass balance stakes by means of automated total stations. The survey dates and coverage are detailed in Table 3. An automated total station is a total station that automatically and quickly follows the target (prism) with a laser while the radial topographic survey is being performed, facilitating the task in the fieldwork. The radial topographic survey is a source-to-receiver procedure, which aims to determine the position of several points over a surface, where the source is a total station with known reference position (XYZ coordinate) and elevation over the ground and the receiver is a prism with known





elevation over the surface. The total station estimates the source-receiver distance and angles in order to calculate the
position of the receiver in relation to the source position. In this study, one single source reference position was used for each
year, which was located outside the glacier area, at a fixed rock position of coordinates E=209143.7, N= 9007853.1, datum
WGS84, UTM18S, elevation: 4764.1 m.a.s.l. The elevation of the total station was estimated with navigator GPS and it was
used the same of each year. Then, each source-receiver surveyed with prism is stored within the memory of the total station.
Once the radial topographic survey has been completed, the data is downloaded and processed.
Topographic data from 2012 and 2014 were surveyed with a TOPCON model GPT 7005L (https://www.al-
top.com/producto/topcon-gpt-7005/), while topographic data and mass balance stakes from 2015, 2017, 2018, 2019 and 2020
were acquired with help of a LEICA TS15 A 3″ R400 (https://surveyequipment.com/assets/index/download/id/844/).
Considering the topographic survey data from 2012 to 2020, the number of points surveyed each year varies from 287 to 342
covering between 11% and 18% of the total glacier area (Table 3).When measuring the distances between nearest points of
every dataset, a mean value of ~25 m was found. In order to obtain the DEMs, the topographic points were converted to a
5m by 5 m raster and interpolated using the Inverse Distance Weighted interpolation tool in ArcGIS.
Due to better resolution and timespan of data, the first dataset was used to estimate the surface elevation changes of the
glacier, by subtracting the digital elevation model of 2020 from that of 2012.
In order to take advantage of the radar data coverage, the bedrock elevation was estimated by subtracting the ice thickness
interpretation from the surface elevation (base on the GPS of the radar survey) in every profile of the GPR data.

**Table 3  Surface elevation dataset used in this study.**

| Date | Format | Number of points | Coverage area as % of total area (5.43 km$^2$) |
|---|---|---|---|
| 2012-09-27 | Points /Raster | 321 | 12 |
| 2014-08-19 | Points /Raster | 287 | 13 |
| 2018-09-06 | Points /Raster | 386 | 18 |
| 2020-09-12 | Points /Raster | 342 | 11 |
| 2014-08-19 | Mass balance stakes (Points) | 14 | 3 |
| 2015-09-01 | Mass balance stakes (Points) | 16 | 3 |
| 2017-05-30 | Mass balance stakes (Points) | 19 | 4 |
| 2018-09-06 | Mass balance stakes (Points) | 18 | 5 |
| 2019-08-20 | Mass balance stakes (Points) | 15 | 4 |



## 2.4 Errors

The estimation of errors in ice thickness ($\varepsilon_{Hdata}$) was obtained following Lapazaran et al. (2016), who split it into two components: a) the error in the value of ice thickness due to GPR measurement ($\varepsilon_{HGPR}$), without taking into account where it was obtained and b) the error in ice thickness due to uncertainties in horizontal positioning ($\varepsilon_{Hxy}$). $\varepsilon_{Hdata}$ is variable and hence, different for every GPR measurement point.

The term $\varepsilon_{HGPR}$ is a function of the propagation velocity in ice (c) and its uncertainty ($\varepsilon_c$), TWTT, and timing error ($\varepsilon_\tau$). The velocity c is dependent of water content and ice purity. In this case, we set it to 0.168 m/ns as used by Gacitúa et al. (2015) in mountain glacier Olivares Alfa in central Andes. Due to lack of measurements of c in tropical Andes, we assume a high percentage of error of 5% in $\varepsilon_c$ as suggested by Lapazaran et al (2016).

The term $\varepsilon_\tau$ is a function of the central frequency of the GPR (see Table 2) and c, which adds up 16.8 m.

The term $\varepsilon_{Hxy}$ should be calculated considering the precision of GPS, speed of transportation and geometric aspects (Lapazaran et al, 2016). $\varepsilon_{Hxy}$ is then calculated a posteriori, unlike the other errors mentioned in this section, because an ice thickness measurement is needed in order to estimate the error that applies on that measurement. In this case, we considered 5 meters of uncertainty in horizontal positioning because of the use of single frequency GPS (there was no dual frequency GPS). With this uncertainty in the XY plane, a 5m by 5m cell point-to-raster for ice thickness procedure was performed, estimating the difference between maximum and minimum interpreted ice thickness within each grid cell and then defining this value as $\varepsilon_{Hxy}/\sqrt{2}$.

An example of the estimated errors in ice thickness $\varepsilon_{HGPR}$, $\varepsilon_{Hxy}$ and $\varepsilon_{Hdata}$ for 2014 dataset is shown in Figure 5. Also, a crossover analysis for the same dataset and for the whole ice thickness dataset is provided in Table 4. The analysis shows a maximum difference of 10 meters, which is less than the radar resolution $\varepsilon_{HGPR}$.
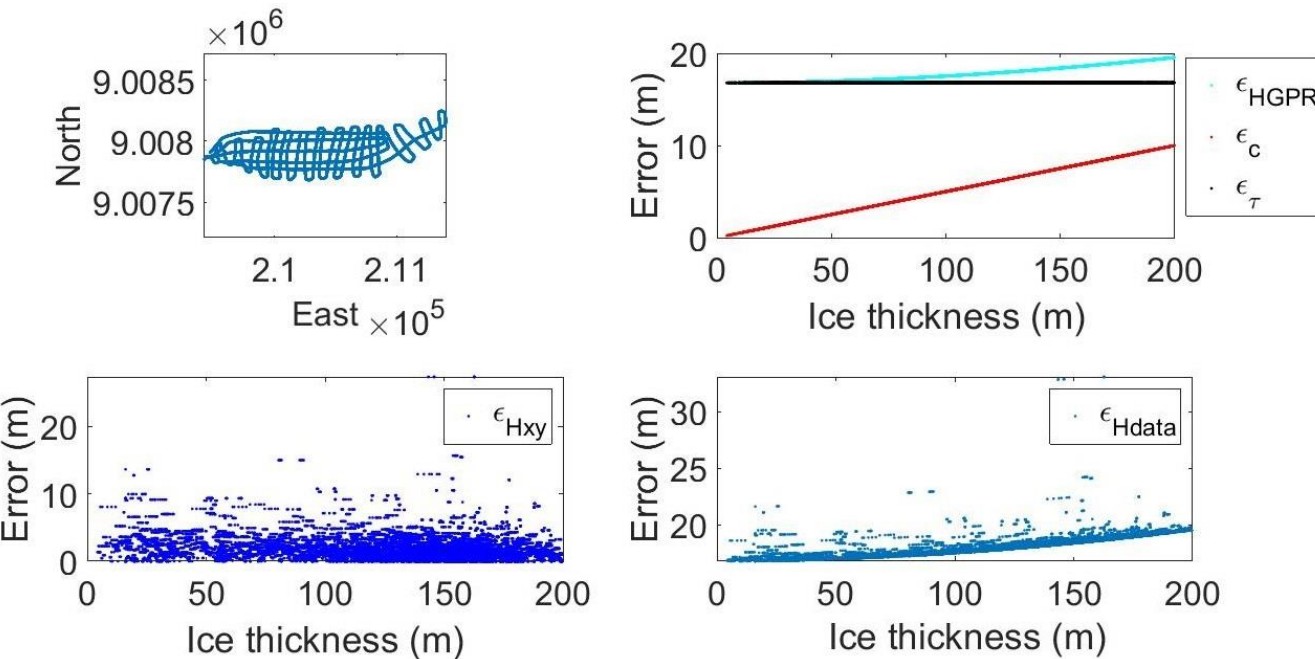

**Figure 5 Estimated errors $\varepsilon_{HGPR}$, $\varepsilon_{Hxy}$ and $\varepsilon_{Hdata}$ for 2014 dataset, calculated according to Lapazaran et al (2016).**

**Table 4  Crossover analysis in ice thickness classified by year.**

| Crossover differences | 2013 | 2014 | 2015 | 2017 | 2018 | 2020 |
|---|---|---|---|---|---|---|
| Number of crossovers | 8 | 109 | 0 | 22 | 0 | 15 |
| Max (m) | 9.4 | 9.3 | No crossovers | 6.5 | No crossovers | 13 |
| Mean (m) | 2.4 | 1.7 | No crossovers | 2.5 | No crossovers | 1.4 |
| Std (m) | 2.9 | 1.5 | No crossovers | 1.9 | No crossovers | 2.6 |
| RMSD (m) | 3.7 | 2.2 | No crossovers | 3.2 | No crossovers | 2.9 |

Regarding the surface elevation obtained by means of total stations, it has an error of ±0.03 m, and then the surface difference estimations have a propagated error of ±0.04 m.

In relation with radar data alone, the GPS elevation has an error of ±15 m (Renfro et al, 2020) and the mean ice thickness error yields ±18 m. Hence, the bedrock elevation has a propagated error of ±23m $\left(\sqrt{15^2 + 18^2}\right)$. On the other hand, the ice thickness differences have a propagated error of ±25.5 m.



In terms of glacier thinning rates, the propagated error is estimated by $\sqrt{\left(\frac{\delta t}{\Delta T}\right)^2 + \left(\frac{\delta y}{\Delta Y}\right)^2}$ where $\delta t$ is the uncertainty in time,
$\Delta T$ is the time period, $\delta y$ is the uncertainty in elevation/thickness difference, and $\Delta Y$ is the elevation/thickness difference,
depending on which term may be used.

## 3 Results and discussion

This study presents the largest record of geophysical prospection using GPR at a glacier in the tropical Andes of Perú. The
overall length of radar profiles adds up 41.2 km covering a total surface of 0.85 km$^2$. The surface elevation of the profiles is
under the ELA, whose mean value between 2014 and 2019 yields 5016 m.a.s.l.

### 3.1 Surface elevation

The glacier surface elevations along profile CC' (Figure 1), which were obtained from interpolated DEMs and mass balance
stakes between 2012 and 2020, are presented in Figure 6(a). The elevation from available stakes measurements are in
agreement with that from DEMs (both data types are found in years 2014 and 2018).

Table 5 Surface elevation differences statistics.

| DEM Difference | Period | | | |
|---|---|---|---|---|
| | 2014-2012 | 2018-2014 | 2020-2018 | 2020-2012 |
| Mean (m) | -5.8 | -12.0 | -7.9 | -25.6 |
| $\delta t/\Delta T$ | 25% | 25% | 25% | 6% |
| Std (m) | 1.3 | 1.9 | 3.0 | 3.6 |
| $\delta y/\Delta Y$ | 22% | 16% | 38% | 14% |
| Prop. Error | 33% | 30% | 46% | 15% |
| Mean/$\Delta T$ (m yr$^{-1}$) | -2.9±1.0 | -3.0±0.9 | -4.0±1.8 | -3.2±0.5 |


In terms of elevation change for the periods 2012-2014, 2014-2018 and 2018-2020, mean differences of -5.8, -12.0 and -7.9
m were found, in the common area of the DEMs which mainly includes the ablation area. This leads to thinning rates of -
2.9±1.0 m yr$^{-1}$, -3.0±0.9 m yr$^{-1}$, and -4.0±1.8 m yr$^{-1}$, respectively (Table 5).
On the other hand, for the whole study period 2012-2020, a mean difference of -25.6 m with a standard deviation of 3.2 m
(13% of the absolute mean value) was found, as shown in Figure 6(b) and in Table 5. Assuming a temporal error of 0.5 years
(6% in the period 2012-2020), the propagated error adds up to 16%, yielding a surface elevation change of -3.2±0.5 m yr$^{-1}$ in
the ablation zone.





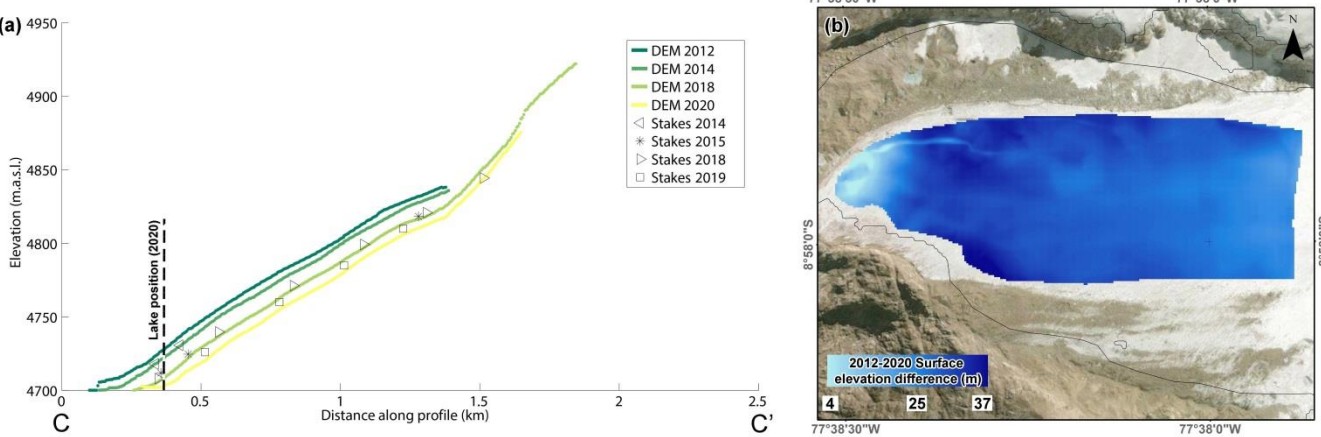

**Figure 6 (a) Glacier surface elevation along profile CC' shown in Figure 1 and (b) surface elevation difference between 2012 and 2020, minimum: 4m, maximum: 37 m, mean difference: 25 m. Background image: Landsat, 4th of July, 2014.**

### 3.2 Ice thickness data

The interpreted ice thickness data at Artesonraju glacier from 2013 to 2020 as well as the 2014-2020 thickness difference are presented in Figure 7, while the main statistics of the interpreted data can be found in Table 6.

The main results show a maximum depth of ~235±18 m (2017), a minimum of ~4±18 m (2014, 2017) and a mean value that varies between 110 m (2020) and 134 m (2013). Data measured in 2018 have not been considered in the summary, due to their less spatial representability in this analysis.

The ice thickness differences between 2014 and 2020 shows a maximum difference of ~45 m and a mean value of ~25m, with 18 m of mean error, yielding an ice thickness change rate of -4.2±3.2 m yr$^{-1}$, which is in agreement with the surface elevation change estimations from section 3.1.

**Table 6  Interpreted ice thickness data set statistics.**

| | Year | | | | | |
|---|---|---|---|---|---|---|
| Ice thickness | 2013 | 2014 | 2015 | 2017 | 2018 | 2020 |
| Mean (m) | 134 | 133 | 129 | 118 | 69 | 110 |
| Std (m) | 1 | 1 | 1 | 1 | 1 | 1 |
| Mean error (m) | 19 | 19 | 19 | 18 | 18 | 18 |
| Min (m) | 6 | 4 | 41 | 4 | 17 | 5 |
| Max (m) | 195 | 200 | 202 | 235 | 112 | 183 |



**Figure 7 Interpreted ice thickness data (raster) from 2013 to 2020 and its error. Background image: Landsat, 4th of July, 2014.**

The surface and bed elevation along profile CC' from 2013 to 2020 by means of radar measurements alone are shown in Figure 8. Also, an interpolated bedrock elevation is provided in Figure 9, using all subglacial elevation available. In the



profile, the bedrock elevation ranges from 4613 m.a.s.l. to 4866 m.a.s.l. The results show a good agreement (within the
magnitude of error) in the bedrock elevation although the surveys are from different years. More evident changes could be
seen at the ablation zone of the glacier or glacier tongue, where the frontal recession is enabling the expansion of the lake
Artesoncocha Alta. Three zones along the CC' profile of Figure 8 can be identified: first, a clear overdeepening zone of the
glacier from the glacier front up to 1km upstream, with a bedrock slope between -5.2° and -3.4°, which can also be observed
in Figure 9; second, a smoothed positive zone, with bedrock slope between 2.4° and 4.4°; and third, a more steep area with
bedrock slope between 14.4° and 16.4°. Similar morphological patterns at the bedrock were obtained by Chisolm (2016),
who estimated bed topography ranging from 4634 m.a.s.l. to 4893 m.a.s.l., indicating appropriate conditions for the
formation of a proglacial lake. The glacier frontal changes of -10.2±2.0 m yr$^{-1}$ (INAIGEM, 2016) and the results obtained in
this study would indicate that in the following years the lake would advance up to the middle point of Artesonraju glacier
tongue, reaching a maximum depth of 82±18 m, estimated as the difference between the current lake surface elevation and
the elevation of the deepest part of the bedrock.

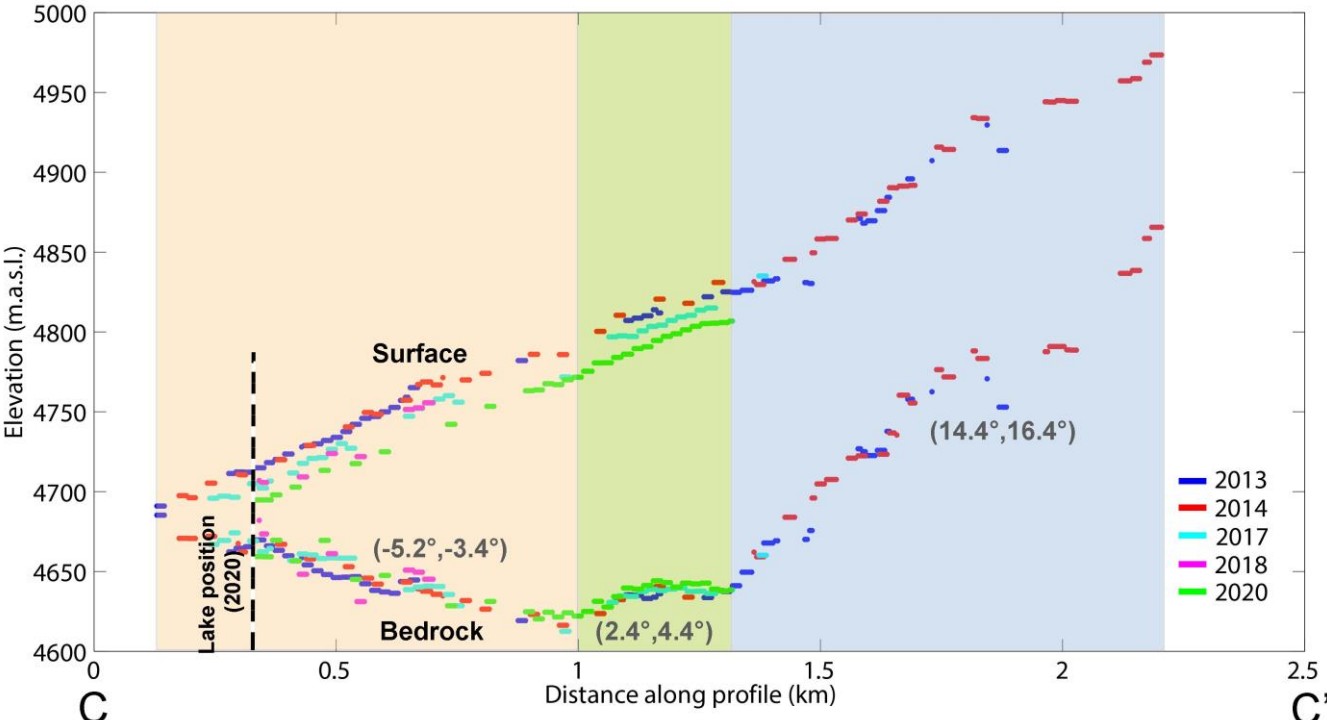

**Figure 8 Surface and bedrock elevation along profile CC' from Figure 1.**

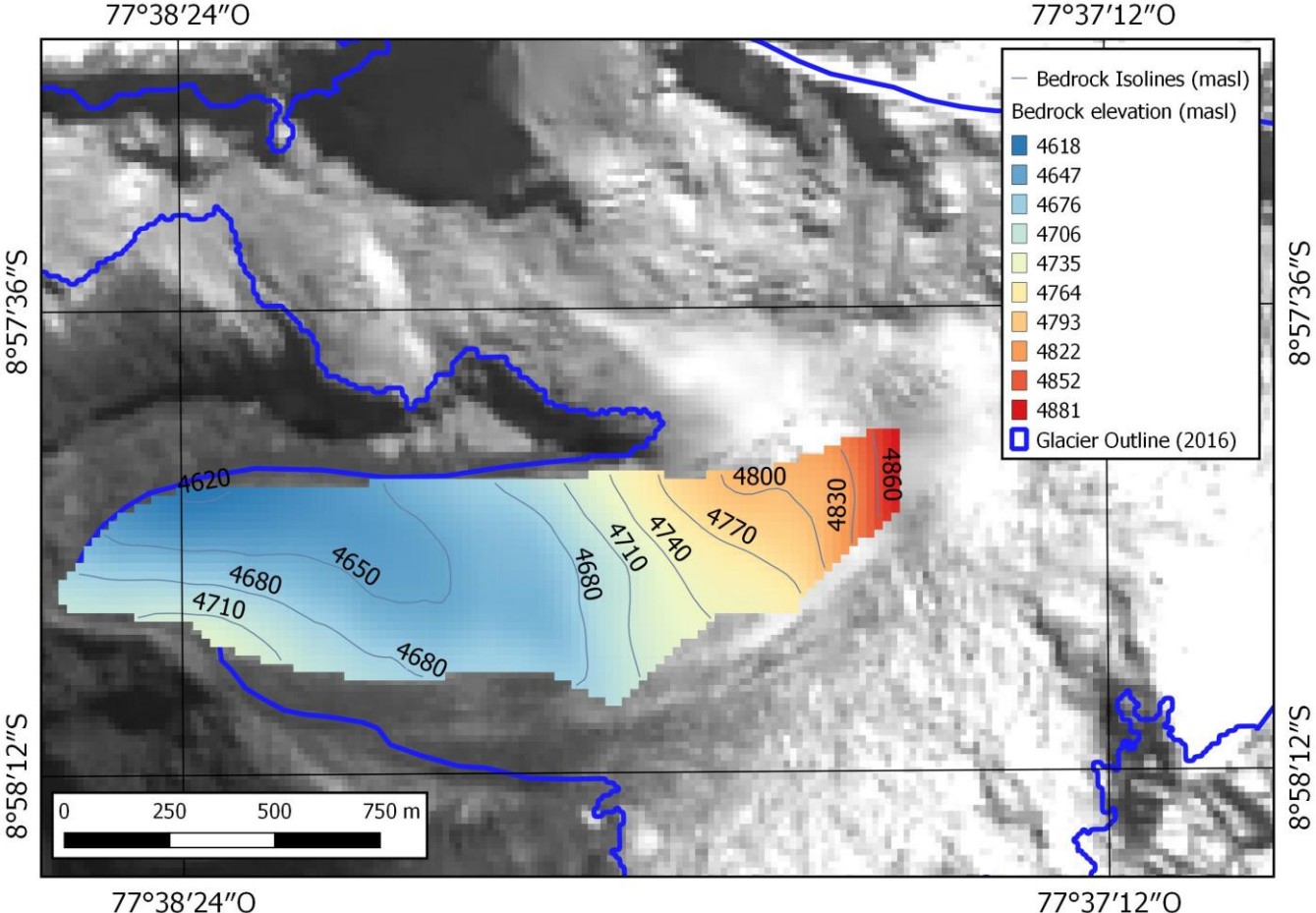

**Figure 9 Bedrock model from all GPR measurements (2013 to 2020). The result was smoothed with a gaussian filter in order to eliminate artifacts that may be generated. Background image: Landsat, 12 July, 2014.**

## 5 Conclusions

We provide a robust dataset contributing with interpreted ice thickness data and ice surface elevation data at the ablation area of Artesonraju glacier in Caraz-Áncash, from 2012 to 2020. Data analysis reveals two aspects: a) a strong decadal mass balance reduction and b) the existence of an overdeepening up to 700 m upstream the central flowline, which may lead to an expansion of lake Artensoncocha Alta up to that point. Data here provided is useful as input for glacier dynamics modelling and can be complemented with other data types for further analysis. It is also necessary to assess the risks associated with the volume increase of the proglacial lake Artesoncocha Alta and with GLOF generation, which may constitute a hazard for the communities located downstream the Artesonraju glacier.



## Data availability

The datasets used in this paper are available at https://doi.org/10.5281/zenodo.5571081   (Oberreuter et al, 2021).

The ice thickness data consists of vector files (*.shape) with the following fields: tr: trace number within the processed profile; x: East coordinates in WGS84/18S, y: North coordinates in WGS84/18S, z: glacier surface elevation (meters above sea level), z: tau_r: two way travel time, depth: ice thickness using 0.168 m/ns, zb: glacier bed elevation (z-depth), e_H_GPR: error in ice thickness due to GPR measurement ($\varepsilon_{HGPR}$), e_H_xyf: error in ice thickness due to horizontal positioning ($\varepsilon_{Hxy}$), e_H_data: total error in ice thickness ($\varepsilon_{Hdata}$).

The surface elevation data from total stations consists of: a) vector files (*.shape) with the following fields: ID, East: East coordinates in WGS84/18S, North: North coordinates in WGS84/18S, H_masl: surface elevation in meters (above sea level), Class: point classification; b) raster files (.tif) with the interpolated surface elevation.

The stakes surface elevation consists of vector files (*.shape) with the following fields: Name: name of the stake, North: North coordinates in WGS84/18S, East: East coordinates in WGS84/18S, H_masl: surface elevation in meters (above sea level).

## Author contributions

INAIGEM and ANA collected the data. JO processed, interpreted the radar data, conception of the work, data analysis and elaboration of figures. EB contributed with the design of the work, data analysis and elaboration of figures, EL contributed with the design of the work and critical revision of the article, KM contributed with the design of the work and critical revision of the article, AC contributed with the design of the work and critical revision of the article, and JU contributed with technical review.

## Competing interests

The authors declare that they have no conflict of interest.

## Acknowledgments

The authors acknowledge financial support from Concytec, the British Embassy, NERC and the Newton-Paulet Fund within the framework of the call E031-2018-01-NERC "Glacier Research Circles", through its executing unit ProCiencia [Contract N°08-2019-FONDECYT] of Peru GROWS project. Also, the authors acknowledge the financial support from the Concytec - World Bank Project "Improvement and Expansion of the National Science Technology and Technological Innovation System Services" 8682-PE, through its executing unit ProCiencia [Contract N°23-2018-FONDECYT-BM-IADT-MU] of Permafrost project.



282

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
