# Peer review of "the ablation area of Artesonraju Glacier, Cordillera Blanca, Perú."

_Earth System Science Data, 2021_

## Referee Comment (RC1)

**Review of "Surface elevation and ice thickness data between 2012 and 2020 at the ablation area of Artesonraju Glacier, Cordillera Blanca, Perú" by Jonathan Oberreuter et al., 2021**

**General comments**

The manuscript and associated data repository provide ice thickness measurements between 2013-2020 over the ablation zone of Artesonraju Glacier, which are used to estimate glacier thinning rates. Additionally, ice surface elevation measurements from automated total stations and stake measurements collected during the same time span are presented. The surface elevation data are gridded into surface DEMs, from which surface lowering rates are estimated. The data is mainly interpreted in terms of glacier thinning rates and potential effects on the proglacial lakes.

This is a comprehensive dataset that, and as the authors point out, could be useful to validate and potentially improve ice thickness modeling efforts. I therefore believe that this dataset is useful for future studies and should be published. The manuscript mostly describes the data with sufficient detail but could benefit from language improvements/sentence structure editing. Overall, I believe that there are several issues that need to be addressed before publication of the manuscript:

- The language of the text is not very precise and could be improved. Often the sentence structure is not very clear which makes it difficult to understand the scientific context.

- The manuscript hints on the effect of thinning and retreat of the Artesonraju Glacier on the formation of the proglacial lake. However, I believe the introduction could be a bit more specific on why changes in this glacier are important to study.

- The section describing the surface elevation dataset is a bit unclear and difficult to understand. It is also unclear to me why the mass balance stakes points were not incorporated into the surface DEMs, as this would provide additional data points for some survey years. Especially, as they appear to be in agreement with the rest of the elevation data and the derived DEMs (L196). Similarly, I wonder if the GPS collected during the GPR surveys could be used as additional data for the DEMs. I suggest comparing the GPS points from the GPR surveys to the other elevation data/DEMs and then based on their agreement/disagreement decide whether to include them or not. Additionally, think it is important to provide a map that shows the point data used to compute the surface DEMs.

- When looking at the 2020-2012 surface elevation difference (Figure 9), there appears to be a band of increased elevation difference in the northern part near the glacier tongue. When looking at the distribution of the measured surface elevation points, this feature aligns with a profile of 2012 data points. Is it possible that this is an interpretation artefact? I suggest adding a brief discussion of this feature in the results section. Additionally, I suggest adding an uncertainty estimate for the interpolated DEMs (see below).

- The Errors section is difficult to understand and needs editing and clarifications (see specific comments below as line-by-line comments). I think this section could be significantly improved just by adding the equations instead of trying to explain them in text format. I also

suggest adding a brief discussion on picking uncertainties, and how unclear bedrock reflections were picked or dealt with. Finally, the errors are based on point data, and not the gridded data. I suggest adding an error estimation for the surface and bedrock DEMs (e.g. by comparing observation points to the interpolated map).

- I appreciate the inclusion of ice thickness changes and a comparison to the ice surface elevation change in the manuscript. However, I think it would be more comparable if the ice thickness data was gridded as well (similar approach as for the ice surface), rather than just comparing crossover point data.

- There is a reflection visible just below the interpreted ice surface in the GPR data. What is this reflection? Is it an artefact or a true interface, for example the snow-ice transition?'

- The Data availability section should be cleaned up such that it matches the data in the zenodo.org repository.

**Specific comments**

L20: "other data types" is a bit vague, I suggest either giving an example, or removing this

L25: (Gonzales Molina & Vacher, 2014) - if possible, please provide a reference in English

L38-39: This is unclear – what association is stronger with lake-terminating glaciers than with land-terminating glaciers? Glacier retreat?

L40: Please clarify which case instead of "one of those cases"

L59: Is it possible to mark the debris-covered part on the glacier? And was this part surveyed in this study?

L71: How was the coverage area calculated, for example from only one GPR profile in 2015?

L74: I was unable to find any information of a fully developed/built GPR system by Unmanned Industrial LTDA. Is it possible that the GPR system was built in-house, and Unmanned Industrial LTDA provided parts for the antenna? Please clarify.

L81: It is unclear what the bandwidth is.

L84: Please specify if this is the range resolution in ice or air, e.g. in brackets.

L90: $c$ is often used for the wave speed in air. To avoid confusion, I suggest using a different variable (e.g. $v$ or $v_{ice}$).

L96-97: I was unable to find information on the RADAR View software. Is this an in-house software? Please specify and provide additional information. And similarly, which GPR company is referred to?

L99: Could you provide an example source of interference that causes signal saturation?

L100-101: Amplitude/max(Amplitude) could be expressed in a better way. E.g. "The data was normalized to the maximum amplitude in each trace."

L115: I suggest adding a reference for the chosen ice velocity here (as has been done in the section below)

L120: When picking the bedrock, were there any challenges? Please add a section on how data was dealt with where the bedrock was difficult to identify

L120: Please specify which GIS software and avoid using the acronym if not defined previously

L133: I suggest specifying how the topographic points were measured, e.g. in brackets

L141: Please provide units for Easting/Northing (I assume meters). Additionally, I believe it would be useful to mark the location of the reference station on the overview map, or in an additional figure that shows the distribution of the elevation point measurements (see comment above)

L152: I suggest replacing "the first dataset" with name/identifier of the dataset used (I am unsure which dataset is referred to here)

L158: To improve this section, I suggest adding equations 2 and 5 from Lapazaran et al, 2016.

L166: Please add the equation used to derive $\varepsilon_\tau$

L171-173: It is unclear how $\varepsilon_{Hxy}$ was derived. After gridding was each radar trace point compared to all values within the grid cell? Or how was $\varepsilon_{Hxy}$ derived for individual radar traces?

L176: $\varepsilon_{HGPR}$ is not equal the range resolution of the radar, it is rather the error associated with the ice thickness

L181-182: How was this calculated/estimated?

L184: Is the mean ice thickness error referring to the mean total error shown in Figure 5 (top right panel)? Please clarify.

L184-185: Please add an equation with variables and specify what each variable is, and then give values. For example, it is unclear where the 23 m comes from in the equation. The same applies to the ice thickness difference error.

L186: Please specify what the uncertainty in time and time period are.

L192: Is it possible to provide a reference for the ELA elevation?

L197: Similar as above, it is unclear how the errors were derived in Table 5.

L216: How were the ice thickness differences computed? Using crossovers within a certain radius?

L227: How was the bedrock DEM computed? On what grid cell size? Please provide this information in section 2.3

L229: "although the surveys are from different years" this is a bit odd, as we don't expect the bedrock to move. But I think one could use this agreement between datasets as argument that the estimated uncertainties are valid.

L230: I suggest replacing "ablation zone of the glacier" with "glacier termini", as most of the survey is in the ablation zone.

L237: Is it possible to provide some sort of estimate on how many years we would expect the lake to advance to the center point of the glacier?

L256: The ice thickness data on the zenodo repository currently are .txt files, not shapefiles.

L256: The current ice thickness data on zenodo don't contain the trace number. I suggest removing this from the text in the manuscript.

L256-266: I suggest providing units in brackets for each parameter described in this section.

L261: Similar to the ice thickness files, the surface elevation data on zenodo are in .txt files, not shapefiles. Additionally, these files don't contain an ID number nor a point classification.

L264: Again, the mass balance stake surface elevations are in .txt files, and the fields do not include the stake name.

**Figures**

- Figure 1: I can't seem to see the Cordillera Blanca (brown) outline on the figure. Please make this outline thicker or add it to the figure if missing.

- Figure 2: Does the black outline represent the glacier outline? Please add this in the figure caption.

- Figure 4: What is the strong reflection below the glacier surface? Is this the ice/snow interface? Please add a brief discussion in the results section.

- Figure 5: I suggest adding labels to the subfigures (a, b, c, d) and describe each panel in the figure legend. The top-left panel is missing units (I assume meter). I suggest making the points in the panel labels showing the colors of individual datasets larger, so they are easier to identify.

- Figure 6: The elevation change in b) appears opposite from the ice thickness change and the discussion. Is it possible the color bar is inverted? I also suggest moving the color bar labels to the top of the bar so it's better readable. Further I suggest adding evenly spaced elevation change labels. Overall, the font size on figure 6 could be increased for better readability.

- Figure 7: I suggest adding figure labels (a, b, c, …), and inverting the order in the top panels (starting from top left to bottom right rather than bottom left to top right). Same as in Figure 6, I suggest moving the labels from within the color bars to the side of the color bars.

- Figure 8: I think this figure provides a good overview of how the ice is thinning and where the proglacial lake could form. I suggest marking the current water level of the lake with a horizontal line.

- Figure 9: I suggest marking profile CC' on this figure

**Quality/completeness of the dataset**

The dataset is available under the given identifier and can easily be downloaded on zenodo.org. Overall, the quality of the dataset is good and the data is easy to use in the current format and size. The metadata includes all necessary descriptions to work with the data. The processing of the GPR data is also appropriate. Ice thickness uncertainties are provided in the dataset .txt files and discussed in the article, although the Errors section of the manuscript requires some clarification and cleanup before publication. Similarly, the Data availability section of the manuscript requires some cleanup to match the data in the zenodo.org repository.

The zenodo.org data archive currently does not include the published bedrock DEM, nor the elevation difference map used in the manuscript. I suggest adding these files to the repository. Additionally, I believe that the GPR data could be useful for other glaciological applications (i.e. changes in subglacial hydrology or englacial water storage between 2013 and 2020). Thus, it would be great to publish the GPR data (raw/processed) along with the interpreted ice thickness dataset).

**Technical corrections**

L12: I suggest deleting "representative"

L14: Add "obtained" in "… surface elevation was obtained by ..."

L15, 175, 202: I suggest replacing "whole" with "entire"

L15: Insert "the" in "The results from the GPR …"

L23: Remove "The" at the beginning of the sentence

L27: Add "the" before "Little Ice Age"

L33: I suggest adding "(mountain ranges)" after cordilleras

L34: largest instead of "larger"?

L36: greatest instead of "greater"?

L48: I suggest changing "ranging normally" to "typically ranging"

L58: Insert "covers" at "… and covers an area…"

L65: Insert "the" in "… while data from the years…"

L76: replace "inbuilt" with "built-in"

L77: delete "shape" (just use bistatic)

L82: does PDA stand for Portable Digital Assistant? I suggest spelling this out instead of using an acronym.

L82: I suggest changing to "For the GPR measurements, a group of five…"

L94: I suggest changing to "b) Picture taken during GPR survey at Artesonraju Glacier"

L97: change to "imported into the Reflexw software…"

L106: space between "2 and"

L112: I suggest changing to "keeping the original horizontal resolution."

L113: replace "real" with "true"

L113: I suggest changing to: "We used a 1D …"

L118: "stored"

L118: "The results enable a correct visualization of the…

L129: add a "the" to "…color cyan and the bedrock in yellow."

L142-143: change to "and the same was used for all survey years."

L145-146: The url link does not work, please update

L152: "Due to the better…"

L155: replace "base" with "based".

L192: replace "under" with "below"

L215: replace "less" with "small"

L233: I suggest to replace "positive zone" with "upward slope"

L235: replace "appropriate" with "favoring" (or something similar)

L247: I suggest deleting "contributing"

---

## Referee Comment (RC2)

[referee-annotated manuscript omitted]

---

## Referee Comment (RC3)

**Surface elevation and ice thickness data between 2012 and 2020 at the ablation area of Artesonraju Glacier, Cordillera Blanca, Perú.**

The manuscript focuses on surface elevation changes and ice thickness measurements in the ablation area of a main tropical glacier at the Cordillera Blanca. This is an interesting topic because of the scarcity of scientific observations reported from this region up to now. Its publication has the potential to be a relevant contribution filling a gap of information from this area. However, it requires a thorough revision by the authors to better present their case.

In general, there is unfortunately a lack of rigor in the writing. All figures could be largely improved, keeping only those that add value to the interpretation of the work done and making sure the captions are fully descriptive of the figures.

In the current state of the manuscript, it is very hard to focus on the details of the data and discuss the results presented here when both the surface elevation and ice thickness methodologies used have been poorly described. Because of the nature of the methods and conditions there is a substantial error/uncertainty intrusion that is attempted to be quantified here, but this is not well described and does not lead the reader to understand the estimations and final accounts.

The conclusions section is unsatisfying because this is a set of 6 years of measurements with good coverage of the ablation area, and although risk assessments or predictions are not within the scope of this manuscript, the authors do not require these series of observations to determine: a) "mass balance reduction" and b) an overdeepening at the base (I expect b) could have been seen from a single data set, for example, from 2014).

**General comments by section:**

L43: This paragraph in particular does not connect well. The introduction needs to be improved.

L55: The study area is poorly described.

L59: Where is the debris covered area? Was it relevant to the GPR observations?

L63: This section needs further description. How was the "coverage area" estimated?

L73: The reference provided for further information (Bello et al., 2020) is not relevant to understand the type of GPR used.

L87 and L88 repeat information.

L90: Use "v" instead of "c".

L95: The GPR data processing lacks a description of the geometrical correction, if any was made. The configuration of the antennas is triangular and depth estimations must consider the antennas geometry. Was it considered for the Migration process?

L131: The method description is unclear. First describe the methods and then the data sets.

L157: The "Error" section is confusing. I suggest adding the equations.

L166: where is 18.8 m coming from?

L176: "The analysis shows a maximum difference of 10 meters"? With reference to Table 4, this is not what is presented.

L176: "...which is less than the radar resolution $\varepsilon_{HGPR}$"?? How was the radar resolution obtained?

L181: How did you get these values?

L183-185 should be better described. Where is ±25.5 m propagated error coming from? And where is it used? Please be more specific. What is "ice thickness differences" referring to?

L202: "...standard deviation of 3.2 m" it does not coincide with the values on Table 5.

L254: The data availability section description doesn't match the data which is referred to.

**Figures:**

Figure 1: Reduce the "Symbology" box. Describe the insets in the caption.

Figure 2: I suggest removing the inset and possibly mark a box in Figure 1 to show the area of Figure 2. Add the position of all surface observations, if possible.

Figure 3: Improve the quality of the text in Figure 3 a). Was it really 10 m separation between the extremes of Tx and Rx?

Figure 4: Are these profiles fully processed? Is "time zero" corrected? Why is there a reflection below the interpreted surface?

Figure 5: The format of this figure could be improved. I suggest: reducing the font size, homogenize the symbols, separate the boxes, make the boxes in equal size and discuss the inclusion of this figure in the text.

Figure 6: This figure is almost not referred to in the text, although it could be a relevant one. Where are the stakes on the maps? Regarding Profile CC': When was it obtained? Why does it appear to cross the lake in Figure 1? Change symbols in a). What was the method used for the interpolation in b).

Figure 7. I don't understand the value of this figure. The order of years is confusing. The color scales are ineffective for the representation. There is no further discussion of this figure in the text.